# Genomic Characterization of *Salmonella typhimurium* DT104 Strains Associated with Cattle and Beef Products

**DOI:** 10.3390/pathogens10050529

**Published:** 2021-04-27

**Authors:** Craig T. Parker, Steven Huynh, Aaron Alexander, Andrew S. Oliver, Kerry K. Cooper

**Affiliations:** 1Produce Safety and Microbiology Research Unit, Western Regional Research Center, Agricultural Research Service, U.S. Department of Agriculture, Albany, CA 94710, USA; craig.parker@usda.gov (C.T.P.); steven.huynh@usda.gov (S.H.); 2Department of Biology, California State University-Northridge, Northridge, CA 91330, USA; aaron.alexander903@my.csun.edu (A.A.); aoliver2@uci.edu (A.S.O.); 3School of Animal and Comparative Biomedical Sciences, University of Arizona, Tucson, AZ 85721, USA

**Keywords:** *Salmonella typhimurium*, DT104, ST19, genomic comparison, beef, cattle, whole genome sequencing

## Abstract

*Salmonella enterica* subsp. *enterica* serovar Typhimurium DT104, a multidrug-resistant phage type, has emerged globally as a major cause of foodborne outbreaks particularly associated with contaminated beef products. In this study, we sequenced three *S.* Typhimurium DT104 strains associated with a 2009 outbreak caused by ground beef, including the outbreak source strain and two clinical strains. The goal of the study was to gain a stronger understanding of the genomics and genomic epidemiology of highly clonal *S. typhimurium* DT104 strains associated with bovine sources. Our study found no single nucleotide polymorphisms (SNPs) between the ground beef source strain and the clinical isolates from the 2009 outbreak. SNP analysis including twelve other *S. typhimurium* strains from bovine and clinical sources, including both DT104 and non-DT104, determined DT104 strains averaged 55.0 SNPs between strains compared to 474.5 SNPs among non-DT104 strains. Phylogenetic analysis separated the DT104 strains from the non-DT104 strains, but strains did not cluster together based on source of isolation even within the DT104 phage type. Pangenome analysis of the strains confirmed previous studies showing that DT104 strains are missing the genes for the allantoin utilization pathway, but this study confirmed that the genes were part of a deletion event and not substituted or disrupted by the insertion of another genomic element. Additionally, cgMLST analysis revealed that DT104 strains with cattle as the source of isolation were quite diverse as a group and did not cluster together, even among strains from the same country. Expansion of the analysis to 775 *S. typhimurium* ST19 strains associated with cattle from North America revealed diversity between strains, not limited to just among DT104 strains, which suggests that the cattle environment is favorable for a diverse group of *S. typhimurium* strains and not just DT104 strains.

## 1. Introduction

*Salmonella* is a major foodborne pathogen, annually resulting in 1.4 million cases in the United States [1] and over 90 million on a global scale [2]. *Salmonella enterica* subsp. *enterica* serovar Typhimurium (*Salmonella typhimurium*) was the most common serovar fifteen years ago, and it is currently the third most common serovar in the United States [3] with many different types of food products, including beef products, associated with outbreaks. In fact, beef products are a critically important vehicle for *S. typhimurium* transmission to humans. Ground beef in particular has been implicated in an estimated 45% of all U.S. *Salmonella* outbreaks during 2002–2011 [4]. Since 1973, there have been 16 *S. typhimurium* outbreaks associated with beef products in the U.S., with ground beef accounting for 37.5% of those outbreaks [4]. While beef consumption in the U.S. is four times higher than the world average [5], this does suggest surveillance of beef products for *S. typhimurium* is critically important to the safety of consumers on a global scale.

*S. typhimurium* is highly clonal, and several types of subtyping methods, including pulsed field gel electrophoresis (PFGE), are not always effective at distinguishing strains [6], which prior to whole genome sequencing (WGS) had made epidemiological investigations more complicated. Several retrospective studies conducted using the higher resolution of WGS have begun to improve the genomic epidemiology of *S. typhimurium*, although most of these studies have concentrated on using SNP analysis to compare the strains [7,8,9,10]. In the examination of genomes from five different outbreaks, one study found that for four outbreaks, epidemiologically linked strain only varied by one or two SNPs. In a fifth outbreak, the epidemiologically linked strains varied by up to 12 SNPs [9]. Whereas, another retrospective study of *S. typhimurium* DT8 strains from an outbreak found only three SNP differences between outbreak-linked strains, but those unrelated to the outbreak had up to 342 SNPs [11]. As the genome epidemiology of *Salmonella* continues to be investigated and expanded, it becomes clear that detailed studies of different species, phage types, sources, and many other variables need to be examined to truly understand the genomic epidemiology of *Salmonella* on a global scale.

*S. typhimurium* phage type DT104 has emerged as a common phage type accounting for up to 50% of all the *S. typhimurium* cases in many countries [12]. DT104 strains have also been linked with numerous outbreaks around the world associated with beef products [13,14,15], including ground beef [16,17]. *S. typhimurium* DT104 is a multidrug-resistant definitive phage type that first emerged in the 1980s [18], and commonly has resistance to ampicillin, chloramphenicol, spectinomycin, streptomycin, sulphonamides, and tetracyclines [19]. Studies have shown that *S. typhimurium* DT104 strains cannot be distinguished by pulsed field gel electrophoresis (PFGE) [20], but multiple locus variable number of tandem repeat analysis (MLVA) has shown the ability to discriminate some strains of DT104 [21]. The advent of whole genome sequencing (WGS) allows for resolution down to the individual nucleotide, which has allowed for expansion of the global genomic epidemiology of DT104 strains. Two major studies have explored the genomics of DT104 strains from different hosts [22] and temporally and spatially different strains [23], but currently no studies have specifically examined DT104 associated with beef products. As cattle and beef products are a critical source of global DT104 infections, as well as *S. typhimurium*, it is vital to gain a better understanding of the global genomic epidemiology.

In 2009, *S. typhimurium* DT104-contaminated ground beef was determined to be the cause of an outbreak in Colorado, USA, leading to 14 clinical cases and the recall of over 466,000 pounds of ground beef [16]. The Colorado Department of Public Health and Environment (CDPHE) isolated several clinical isolates and an additional isolate from a ground beef patty obtained from the home of a patient. The goals of this study were to (1) sequence and compare two clinical isolates and ground beef source strains; (2) conduct a detailed genomic comparison of *S. typhimurium* clinical and bovine-associated DT104 and non-DT104 strains; (3) compare the ground beef outbreak strains against a global DT104 population; (4) compare these bovine-associated DT104 strains to other bovine-associated *S. typhimurium* from North America, which will provide a detailed genomic characterization of DT104 strains associated with cattle and beef products.

## 2. Results

### 2.1. 2009 Colorado Ground Beef Outbreak DT104 Strains

The complete genome of *S. typhimurium* DT104 strain RM9437 (ground beef) from the 2009 Colorado outbreak is 5,030,429 bp (composed of 4,936,499 bp chromosome and 93,930 bp plasmid), and encodes for 4569 CDSs, 7 rRNA operons, and 83 tRNAs. The genome contains eight prophages, including six complete and two incomplete prophages (Table 1). The plasmid pRM9437 is >99% identical to pLT2, the described virulence plasmid found in many *S. typhimurium* strains. It possesses *spvRABCD* and *pefBACD* (fimbriae) and *srgABC* (involved in fimbriae biogenesis). This version of the plasmid does not possess antibiotic resistance genes. The strain also possesses the *Salmonella* Genomic Island 1 (SGI1), which is the multidrug resistance element possessing resistance genes to ampicillin, chloramphenicol, streptomycin, sulfonamides, and tetracycline (ACSSuT resistance) on the chromosome and described previously [24,25]. Based on the sequences of seven housekeeping genes used for multilocus sequence typing (MLST) of *S. enterica*, the sequence type for these outbreak strains was determined to be ST19. Two clinical strains from the outbreak, RM9435 (clinical) and RM9436 (clinical), were used for manual high-quality SNP curation between clinical strains and ground beef source strain from the same outbreak, which revealed that there are no SNPs between these three strains. Therefore, all additional SNP analysis only involved RM9437.

### 2.2. SNP Analysis between S. typhimurium DT104 and Non-DT104 Strains

Thirteen *S. typhimurium* ST19 strains with complete genomes were selected for further SNP analysis (Table 1). These strains included seven from bovine sources (three DT104 and four non-DT104) and six from clinical sources (four DT104 and two non-DT104). Using CSI Phylogeny SNP analysis software and RM9437 as the reference genome, we determined that the seven DT104 strains had significantly less SNPs compared to the six non-DT104 strains (Figure 1). This finding was strengthened by performing SNP analysis of all the strains against each other, which found a total of 1781 high-quality SNPs between all strains with an average of 440.9 SNPs between any two strains. However, the DT104 strains possessed an average of 55.0 SNPs compared to 474.5 SNPs for non-DT104 strains (Table 2). Phylogenetic analysis of the thirteen ST19 strains reinforced this separation of the DT104 and non-DT104 strains, as all of the DT104 strains clustered together, and demonstrated a common ancestor deviation from non-DT104 strains (Appendix A). Yet the source of isolation did not have a similar impact on the relationship of the *S. typhimurium* strains regardless of the phage type, as neither bovine nor clinical strains clustered together on the maximum-likelihood (ML) tree even within the separate phage type clusters. While it would be expected that clinical strains would not cluster together unless they had a common infectious host source, it was interesting that the cattle strains did not cluster together, which suggests there is diversity among *S. typhimurium* strains associated with cattle. Furthermore, the bovine isolates did have significantly more SNPs between the strains than the clinical strains, as the bovine strains averaged 511.8 SNPs compared to an average of 344.9 SNPs for the clinical strains (Table 2).

### 2.3. Gene Content Analysis of S. typhimurium DT104 and Non-DT104 Strains

Core genome analysis of the thirteen strains found that there were 4300 core genes, 877 shell genes (present in 1 ≤ strains > 12), and 587 cloud genes (present in only one strain) among all strains, which was fairly consistent for non-DT104 strains (4331 core genes), bovine strains (4308 core genes), and clinical strains (4350 core genes). However, DT104 strains as a group had appreciably more core genes at 4512 compared to these other groups, which also resulted in fewer shell genes (300) and cloud genes (49) compared to the rest of the overall group (Appendix A). Fifty-three (53) genes unique to all the DT104 strains were identified in the pan-GWAS analysis, which were composed entirely of genes from prophages, although based on the PHASTER analysis there was some variation in the specific prophage that contained the genes between the different DT104 strains. For example, the 53 DT104 unique genes in *S. typhimurium* strain DT104 included 38 genes that were present in an intact *Aeromonas* phage vB AsaM-56 (as identified by PHASTER analysis), and the other 15 genes were present in an intact *Salmonella* phage ST64B (as identified by PHASTER analysis; Appendix A), whereas the 53 unique genes in *S. typhimurium* str. RM9437 were broken down as 38 genes in an intact *Edwardsiella* phage GF-2 (as identified by PHASTER) and the other 15 genes were present in an intact *Salmonella* phage 118970 sal3 (as identified by PHASTER). The difference in identified intact prophages containing the DT104 unique genes is likely due to PHASTER analysis, but could be worth investigating in the future. On the other hand, none of the 12 genes found to be unique to the non-DT104 strains or missing from DT104 strains were associated with prophages, and the genes included 11 genes in the allantoin–glyoxylate metabolic pathway and one hypothetical protein that is 87 amino acids long (Appendix A). These 11 allantoin–glyoxylate metabolic pathway genes seem to have been deleted in the DT104 strains as there is nothing inserted or replacing the genes in the genomes (Appendix A). No unique genes were identified for either the bovine strains or the clinical strains during the pan-GWAS analysis.

### 2.4. Prophage and Antibiotic Resistance Analysis

Phage type DT104 strains are considered a significant public health concern as they are frequently multidrug resistant, therefore the intact prophage profile and antibiotic resistance gene profile for all the strains were examined. Similar to the SNP analysis, DT104 and non-DT104 strains were found to have very different profiles for both the prophages and antibiotic resistance genes (Figure 2). Yet, the phage Gifsy-2 was common to all of the strains in the study and 11/13 (84.6%) possessed the phage Gifsy-1. Additionally, 5/7 (71.4%) of the DT104 strains contained the *Edwardsiella* phage GF-2 (defined by PHASTER), but none of the non-DT104 strains had this prophage. The *Enterobacteria* phage ST104, which is commonly present in DT104 strains [28], was also found in 6/7 (85.7%) of the DT104 strains. Interestingly, prophage ST104 was completely missing (not incomplete or questionable either) from the DT104 strain CDC 2011K-1702, but the prophage was present in the non-DT104 strain CDC 2010K-1587. Further analysis shows that CDC 2011K-1702 has the prophage *Enterobacteria* phage P4 inserted at the site of *Enterobacteria* phage ST104 insertion in the other DT104 strains. Nucleotide alignment of the prophage ST104 sequence from CDC 2010K-1587 (non-DT104) against the prophage sequences from DT104 strains (Appendix A) showed only 67.44% identity. Alignment of the prophage ST104 sequences from DT104 strains against each other showed 99.99% identity. Bovine strains also had quite a bit of prophage variation, as none of the bovine strains shared an identical prophage pattern. There were 13 different prophages identified between the seven bovine strains, but only two prophages were shared by all the strains. The other 11 prophages were present in at less than 50% of the strains. None of the clinical strains had identical prophage patterns. The six clinical strains had a total of 10 different prophages identified, but only the phage Gifsy-2 was present in all the clinical strains.

While there was some diversity in the antibiotic resistance genes present in the different *S. typhimurium* genomes, it was not as distinctive as the prophage diversity among the strains. In fact, all 13 of the strains contained the *aac(6′)-laa* gene that encodes for aminoglycoside resistance (Figure 2). According to the Comprehensive Antibiotic Resistance Database (CARD) [29], this gene is present in 91.3% of *Salmonella enterica* genomes. However, there were some differences in antibiotic resistance genes between DT104 strains and non-DT104 strains. For example, none of the non-DT104 strains contained the previously described SGI1 encoding ACSSuT resistance. For the DT104 strains, 5/7 (71.4%) had the entire genomic island. As a group, the bovine strains did not share an antibiotic profile. This was also the case for the clinical strains.

### 2.5. Global Genomic Epidemiology of DT104 Strains

We performed hierarchical clustering (HC) of the core genome multilocus sequence types (cgMLST), which provides a relationship between the three Colorado outbreak strains (RM9435, RM9436, and RM9437) and a global set of DT104 strains at different HC levels. At the hierarchal clustering 100 (HC100) level, the differences between strains’ cgMLST alleles are 100 or fewer, and this level has been associated with strains from long-term endemic persistence [30]. At this level, DT104 strains from bovine, swine, and food sources were distributed across the minimum spanning tree (MST) based on the cgMLST. These strains were also intermixed with human clinical strains across the entire MST, whereas DT104 strains from avian sources formed a specific cluster with only a few clinical strains and a single bovine strain. The U.S. outbreak strains clustered together as an individual node and away from most other bovine DT104 strains, but the clinical and ground beef forming a single node confirms the outbreak relationship of these strains. The country of origin does not seem to impact the clustering of the DT104 strains even for similar sources of isolation (Figure 3). To examine the relationship of these global DT104 strains a little closer, strains RM9435, RM9436, and RM9437, 25 bovine strains, and a random selection of 72 strains representing other sources of isolation were selected for pangenome analysis using Anvi’o software. The analysis found 4362 core gene clusters, 621 accessory gene clusters, and 424 singleton gene clusters among all 100 strains. Similar to cgMLST analysis, when clustering the strains based on presence/absence of the gene clusters, the swine and bovine strains did not all cluster together and were intermixed with the clinical strains (Figure 4). Examination of genes present or absent from DT104 strains associated with cattle found that there were no genes present in only cattle-associated DT104 strains and absent in non-cattle DT104 strains or vice versa. However, there were 11 genes that were statistically significant in presence or absence from cattle-associated DT104 strains when compared to non-cattle-associated strains. Only two genes were more commonly present in cattle DT104 strains than in non-cattle-associated DT104 strains. The beta-lactamase PSE-1 (*pse1*) gene was present in all 26 bovine DT104 strains examined, but only present in 75.0% of the 72 non-cattle-associated DT104 strains (*p*-value = 0.0026). The tRNA-Ala gene was present in 19.2% of cattle-associated DT104 strains compared to only 1.4% of non-cattle-associated DT104 strains (*p*-value = 0.0047). Whereas, the other nine genes were statistically more likely to be missing from cattle-associated DT104 strains than from non-cattle DT104 strains, including five hypothetical genes encoding proteins, the regulatory protein Rop, mobilization protein MbeC, endoribonuclease ToxN, and dihydropteroate synthase (Table 3).

### 2.6. Genomic Epidemiology of Cattle-Associated S. typhimurium Strains

To further examine the genomic epidemiology of bovine-associated *S. typhimurium* strains, the fifteen ST19 *S. typhimurium* strains were compared to both bovine-associated DT104 and non-DT104 ST19 *S. typhimurium* strains from North America. The MST based on the cgMLST analysis demonstrated quite a bit of diversity among the various bovine strains as there were not many large nodes. However, this analysis did separate the fifteen DT104 and non-DT104 strains from each other. All DT104 strains branched off the same major node and away from six non-DT104 strains (Figure 5). The four non-DT104 bovine strains were also spread nearly across the entire MST. Overall, the results suggest that *S. typhimurium* strains associated with different bovine sources are extremely diverse even within a single geographical region, particularly as *S. typhimurium* is a highly clonal group as a whole.

## 3. Discussion

*S. typhimurium* strains are highly clonal, and many traditional subtyping methods such as pulsed field gel electrophoresis (PFGE) have trouble resolving strains during an outbreak [6]. This was particularly true with DT104 strains, as prior to the advent of whole genome sequencing (WGS) only MLVA was shown to have some resolution to differentiate DT104 strains [21,31]. Using WGS, this study investigated the level of clonality of DT104 strains within an outbreak, compared bovine and clinical DT104 strains to non-DT104 strains from within the same sequence type (ST19), additional DT104 strains from around the world, and cattle-associated *S. typhimurium* strains from North America. *S. typhimurium* DT104 strains are a common cause of disease in many parts of the world, and improving the genomic epidemiology of this phage type of *S. typhimurium* provides input information for public health laboratories and clarifies the role cattle and beef products may play in human infection. The SNP analysis between source and clinical strains from an outbreak revealed no SNP differences, which was confirmed at the allele level via cgMLST analysis. This suggests that due to the high level of clonality, there will be a limited number of SNPs between clinical DT104 strains during an outbreak, but further outbreak investigations are needed to confirm these results. However, this information can be used for improving source attribution of DT104 strains, and ultimately to improve source-tracking to assist in rapidly conducting future outbreak investigations to minimize the public’s exposure to contaminated food products.

The advent of WGS has dramatically improved the ability of public health agencies around the world to source track foodborne outbreaks [32]. Nevertheless, the continued development and improvement of WGS as a critical public health tool for outbreak source tracking for pathogens is vital to improving global food safety, particularly understanding SNP variation between different serovars and phage types of *Salmonella*. SNP analysis of the bovine and clinical strains within the ST19 group averaged 440.9 SNPs. That average was slightly higher than was found in another study (380.6 SNPs) [33]; however, that study examined fewer strains than our study. Furthermore, our study found that DT104 strains had significantly less SNPs than non-DT104 ST19 strains as a whole, with a range of 30 to 85 and an average of 55.0 SNPs between strains. Non-DT104 strains had a range of 10 to 646 and an average of 474.5 SNPs. Mather et al. found that out of 359 DT104 strains, there was a range of 0 to 167 SNPs between the strains [22], which even on the high end of the range is significantly lower than that of most non-DT104 strains from the ST19 group. Taken together, these data provide genomic evidence that DT104 strains have a higher level of clonality when compared with other ST19 *S. typhimurium* strains. Thus, they may require special consideration during analysis for outbreak source tracking.

The detailed analysis of DT104 strains from bovine sources and clinical sources against non-DT104 strains found that not all DT104 strains contained the SGI1 genomic island that encodes multidrug resistance, which has been described before for certain DT104 strains. For example, DT104 strains have been described before with a partial deletion of SGI1 that lacked *flo* and *tet* genes [34], which is similar to the DT104 strain CDC 2011K-1702 that was used in this study. Although, we also found that DT104 strain SA972816 from China was missing the entire 13 kb multidrug resistance section of the SGI1 genomic island, which is not as common in DT104 strains, but has been described previously [23,35]. The pangenome analysis using Roary/Scoary of the 98 DT104 strains to determine DT104 cattle unique genes found that all the cattle-associated DT104 strains (DT104 cattle-associated strain SA972816 was not included in this particular analysis) had the beta-lactamase PSE-1, but it was missing in 25.0% of the non-cattle-associated DT104 strains. The *pse1* or *blaP1* gene is in the SGI1 genomic island and encodes for ampicillin resistance [36], which suggests that 25.0% of the non-cattle-associated DT104 strains could be missing part if not all of the SGI1 genomic island. However, it seems much more common for cattle-associated DT104 strains to have the SGI1 genomic island that encodes for multidrug resistance, which might be due to the heavy use of antimicrobials in the beef industry, but would need further confirmation. We investigated the *Enterobacteria* phage ST104 from several DT104 strains as this is the phage that is believed to provide the DT104 phage type [28], and it remains unclear as to how DT104 strains CDC 2011K-1702 could be missing the phage and still be a DT104 strain. Several possibilities exist: (1) it is not a DT104 strain and was mis-identified by the CDC, but it did group with the other DT104 strains in the phylogenetic analysis although it was also missing the SGI1 genomic island; (2) multiple bacteriophage that are closely related could be responsible for the DT104 phage type. Currently, additional analysis is needed to determine the answer to this issue. However, the reason for the non-DT104 strain carrying the ST104 phage was revealed to be that there was only ~67% sequence homology, which suggests the phage may not be functional in this strain, thus impacting the phage type.

It has previously been described that the allantoin utilization loci is deleted in DT104 strains [37,38]. Yet, the previous types of analysis were not able to determine if the genes were disrupted due to insertion, deletion, or substitution in the DT104 genomes. Therefore, this study is the first to show that the allantoin utilization loci was deleted from the DT104 genome, which suggests it might not be needed for DT104 strains during colonization of common isolation sources like cattle and swine. In fact, Matiasovicova et al. investigated if deletion of allantoin utilization loci in *S. typhimurium* affected virulence in mice or colonization in chickens, since poultry have high levels of allantoin. The study found that the loss of the loci *S. typhimurium* does not appear to affect virulence in mice or colonization of chickens [38], so this may not be the reason for the rarity of DT104 strains in poultry. Thus, the reason for the deletion of the allantoin utilization loci in DT104 strains, whether in colonization, survival, virulence, or for another reason, still needs further investigation.

DT104 strains are commonly isolated from swine and cattle, yet not as frequently from poultry [39], and the results of this study demonstrate that it could be due to more diversity in the DT104 strains associated with cattle and swine compared to those associated with poultry. The cgMLST analysis of DT104 strains from EnteroBase found that those strains isolated from poultry were clustered together compared to cattle and swine strains. Cattle and swine strains were distributed across the MST, suggesting a significant amount of diversity among these strains. It should be noted that all of the poultry strains were isolated in Ireland, which could explain the lack of diversity in the poultry-associated strains. However, cattle strains as strains restricted to specific European regions (Germany, Ireland, and Northern Ireland) were spread across the tree and not clustered together by country. The diversity observed in this study was at the HC100 level that is associated with long-term endemic persistence in a host [30], therefore it suggests that cattle and swine are endemically colonized with DT104 strains for long time periods, which results in more frequent isolation and higher risk of human infection. Overall, we suggest that DT104 strains have established a niche in cattle and swine populations around the world that has resulted in numerous outbreaks associated particularly with contaminated beef products. Interestingly, analysis of DT104 strains to determine unique genes associated with cattle colonization found that there were no unique genes associated only with DT104 cattle strains, which suggests that DT104 strains colonizing cattle are more generalist and not specialist, thus probably resulting in increased diversity of the population associated with cattle across the world. Furthermore, when ST19 lineage *S. typhimurium* strains associated with cattle, including multiple DT104 strains, from a defined region (North America) were examined using cgMLST, there was also quite a bit of diversity in general among *S. typhimurium* strains. Additionally, the diversity of the cattle-associated DT104 strains was further supported as the multiple DT104 cattle strains were spread across the tree and not together in a tight cluster, even compared to non-DT104 strains.

In conclusion, this study was the first to explore the genomics of DT104 strains strictly associated with cattle and/or beef products from the perspective of understanding the genomics and genomic epidemiology for improving global food safety. The results of the study found that DT104 strains are more clonal than non-DT104 *S. typhimurium* strains even from the same ST, but this changes when the DT104 strains are associated with cattle. In fact, an association with cattle and/or beef products seems to increase the diversity of *S. typhimurium* strains in general even within limited geographical areas. Therefore, from a global food safety perspective, it should be understood that DT104 and non-DT104 *S. typhimurium* strains associated with cattle and/or beef products can be quite diverse. However, DT104 strains from outbreaks should have clinical strains and source strains with practically no SNPs, like that demonstrated for the 2009 Colorado outbreak strains in this study.

## 4. Materials and Methods

### 4.1. Strains

All *S. typhimurium* strains sequenced (RM9435, RM9436, and RM9437) and/or used for the detailed comparative genomics or SNP analysis in this study are listed in Table 1 along with their respective accession number [22,26,27]. All strains sequenced in this study were grown on LB agar and/or LB broth at 37 °C unless otherwise stated and were sequenced as described below. Additionally, all global *S. typhimurium* strains utilized for cgMLST analysis in EnteroBase (http://enterobase.warwick.ac.uk/; accessed 11 February 2021) [30,40] including DT104 and North American bovine-associated strains were available on 11 February 2021 and 14 May 2020, respectively. Additionally, those strains listed above and the 100 DT104 strains selected for pangenome/DT104 cattle-specific analysis are also listed in Appendix A.

### 4.2. DNA Extraction

Genomic DNA for the three *S. typhimurium* strains (RM9435, RM9436, and RM9437) that were sequenced in this study were prepared from single colonies grown overnight in LB broth, and then extracted as previously described [41]. Briefly, bacteria were lysed with lysozyme (20 mg/mL in 50 mM Tris, pH 8.0; Fisher Scientific) and 10% SDS and then sequentially treated with RNase A (1 mg/mL at 37 °C for 24 h; Fisher Scientific, Waltham, MA, USA) and proteinase K (20 mg/mL at 37 °C for 4 h; Fisher BioReagents, Waltham, MA, USA). Extracted DNA was then precipitated with a sodium acetate/ethanol solution, purified by phenol/chloroform extraction, precipitated again with ethanol, and finally, re-suspended in Buffer EB (Qiagen; Hilden, Germany) for DNA sequencing library preparation. Prior to DNA library preparation, the DNA concentration was determined using a Qubit 3 fluorometer (Thermo-Scientific; Waltham, MA, USA) measuring at 485/530 nm and quality-assessed using a Nanodrop ND-1000 spectrophotometer (Thermo-Scientific; Waltham, MA, USA) measuring at 260/280 and 260/230 ratio wavelengths.

### 4.3. Genome Sequencing

DNA sequencing libraries were prepared using the KAPA Low-Throughput Library Preparation Kit with Standard PCR Amplification Module (Kapa Biosystems; Wilmington, MA, USA) following the manufacturer’s instructions except for the following modifications: 750 ng of DNA was sheared at 30 psi for 40 s and size selected to 700–770 bp following Illumina MiSeq protocols. Standard desalted TruSeq LT and PCR primers (Integrated DNA Technologies (IDT); Coralville, IA, USA) were used at 0.375 and 0.5 µM final concentrations, respectively. The PCR reaction was reduced to 3–5 cycles. Libraries were quantified using the KAPA Library Quantification Kit (Kapa Biosystems; Wilmington, MA, USA), except using a 10 µL volume and 90 s annealing/extension PCR, then pooled and normalized to 4 nM. Pooled libraries were re-quantified by digital droplet PCR (ddPCR) on a QX200 system (Bio-Rad; Hercules, CA, USA), using the Illumina TruSeq ddPCR Library Quantification Kit and following manufacturer’s protocols, except with an additional 2 min annealing/extension time. All libraries were sequenced on an Illumina MiSeq instrument (500-cycle v2 sequencing kit; Illumina; San Diego, CA, USA) following the manufacturer’s protocols. Sequencing resulted in a total of 2,477,178 paired-end (PE) 250 bp reads for RM9435 (~137× coverage); 1,265,954 PE 250 bp reads for RM9436 (~70× coverage), and 1,290,788 PE 250 bp reads for RM9437 (~71× coverage).

### 4.4. Assembly and Annotation

The PE sequence reads for *S. typhimurium* str. RM9437 were assembled de novo using the Roche Newbler assembler (v2.3) using default parameters, which resulted in a total of 70 contigs. Next, the 70 contigs for RM9437 were ordered by aligning against *S. typhimurium* str. DT104 (HF937208.1 and HF937209.1) using Mauve software [42]. The genome was closed in silico using the following steps: (1) identification of repeated contigs using the Perlscript Contig_extender3 [43] and determination of the number of reads per contigs length, and (2) contig extension using Illumina reads and Newbler contigs within Geneious software (v9.1) [44], which resulted in a single chromosome and a single plasmid sequence for *S. typhimurium* str. RM9437.

All coding DNA sequences (CDSs) were determined using the Glimmer 3.0 [45] plugin for Geneious software, and then predicted CDSs were compared against annotations for *S. typhimurium* LT2 genome (AE006468.1) and *S. typhimurium* DT104 (HF937208.1). Novel or unclear annotations were identified by using BLASTP against the National Center for Biotechnology Information (NCBI) nonredundant (nr) database with the parameters ≥80% identity across ≥75% of the query sequence for a confirmation hit. All tRNAs and rRNA loci were identified by using BLASTN again the *S. typhimurium* str. LT2 genome, and the final annotations were confirmed by comparing against the automated NCBI Prokaryotic Genome Annotation Pipeline (PGAP) annotation [46]. Reads for strains RM9435 and RM9436 were individually assembled in EnteroBase automatically upon submission, and both strains were only used for core genome multilocus sequence typing (cgMLST) and SNP analysis against RM9437 in the study.

### 4.5. Single Nucleotide Polymorphism (SNP) Analysis

SNP variations between *S. typhimurium* strains RM9435, RM9436, and RM9437 were identified by mapping the RM9435 or RM9436 reads against the reference genome of *S. typhimurium* str. RM9437 using Geneious Prime 2020.1.2 software. Variation/SNPs were found using the following parameters: minimum 30× coverage, >80% of reads contain variation, maximum variant *p*-value of 10^−6^, and minimum strand bias *p*-value of 10^−5^. Repeat regions were identified by BLASTN search of all SNP regions against the RM9437 chromosome. Those with multiple hits (≥50% identity across ≥50% of the query sequence) were examined to identify repeat regions, and any additional repeat regions were identified using the RepeatFinder plugin in Geneious Prime 2012.1.2 software using the following parameters: 100 bp minimum repeat length and 1% maximum mismatches. All repeat regions were then removed from the results of the SNP analysis.

The SNP matrix of the twelve complete genomes of the ST19 *S. typhimurium* strains lacking sequence reads was generated using CSI Phylogeny 1.4 [47] (https://cge.cbs.dtu.dk/services/CSIPhylogeny/; accessed on 14 May 2020) from the Center for Genomic Epidemiology. The complete genome sequence of RM9437 was included in the analysis as a reference genome to generate the SNP matrix, using the following parameters: 30× minimum depth at SNP positions; 25% minimum relative depth at SNP positions; 100 bp minimum distance between SNPs (pruning); 50 minimum SNP quality; 25 minimum read mapping quality; 1.96 minimum Z-score.

SNP analysis was also conducted among the 13 different bovine and clinical *S. typhimurium* strains using Parsnp software (v1.2) [48], and the SNP density of strains visualized using Gingr software (v1.2) as part of the Harvest suite [48]. A maximum-likelihood tree was generated by using the 1781 filtered high-quality SNPs identified by the program, which were concatenated and aligned using command line MUSCLE software (v3.8.31) with the default parameters [49,50], and a maximum-likelihood (ML) tree generated with MEGAX software (v10.1.8) [51]. First, a best-fit model was determined for the alignment data, then a maximum-likelihood (ML) tree was generated using the Tamura-Nei model [52] with Gamma distribution with 1000 pseudoreplicates.

Identified SNPs of the thirteen different ST19 *S. typhimurium* strains by either Parsnp or CSI Phylogeny are listed in Appendix A. A SNP map demonstrating the SNP differences among thirteen strains using the identified SNPs from CSI Phylogeny and *S. typhimurium* str. RM9437 as the reference genome was generated using BLAST Ring Image Generator (BRIG) software [53]. Locations of the forward and reverse coding DNA sequences (CDSs), prophage, and repeat regions in the RM9437 genome were also included for reference to location of SNPs.

### 4.6. Prophage and Antibiotic Resistance Genes

Prophage sequences in all 13 of the strains used in the detailed comparative genomic analysis (Table 1) were identified using the online tool PHASTER (www.phaster.ca; accessed 11 February, 2021) [54]. To compare the identified *Enterobacteria* phage ST104 prophage sequences of the non-DT104 strain CDC 2010K-1587 and the six DT104 strains that contain the prophage, the sequences were aligned using the MUSCLE plugin with default parameters and 10 iterations in Geneious Prime 2020.1.2 software. Whereas, antibiotic resistance genes present in the genome of the different *S. typhimurium* strains were identified using the online tool ResFinder (v4.0) [55] from the Center for Genomic Epidemiology.

### 4.7. Allele-Based Core Genome Multilocus Sequence Typing (cgMLST) Analysis

Allele-based genomic epidemiology analysis of the three Colorado DT104 outbreak strains (RM9437, RM9436, and RM9435) were compared to other DT104 strains from around the world, whereas the other twelve *S. typhimurium* DT104 and non-DT104 strains from bovine and clinical sources were compared to bovine-associated ST19 strains publicly available on EnteroBase on 11 February, 2021 or 14 May, 2020, respectively. An MST utilizing the improved minimal spanning tree algorithm MSTree V2 of GrapeTree [56] based on cgMLST allele differences of 2978 alleles was generated using the three DT104 Colorado outbreak strains sequenced for this study and 556 DT104 *S. typhimurium* strains comprising 330 different STs in EnteroBase at a hierarchal clustering level 100 (HC100). The cgMLST scheme utilized in Enterobase that was employed in this study has been previously described [30]. MST based on cgMLST allele differences was also generated using the fifteen ST19 strains and 775 ST19 *S. typhimurium* strains with 682 STs isolated from different bovine-associated sources in North America.

### 4.8. Genes Unique to DT104 and Cattle DT104 Strains Analysis

Genome-wide association studies (pan-GWAS) analysis was conducted on the thirteen ST19 *S. typhimurium* genomes (Table 1). First, each genome was annotated using Prokka (v1.13.3) [57] to the bacteria kingdom using default parameters. Next, the pan genome at 90% protein identity was determined for the thirteen strains using the program Roary (v3.12.0) [58] with the following options: create a multiFASTA alignment of core genes using PRANK and fast core gene alignment with MAFFT (Roary command: roary -f./DT104 -e -n -v -i 90./DT104/*.gff). The number of genes in the core genome, soft-core genome, cloud genes, and shell genes that were output from Roary was visualized using Roary.plots.py program. The entire Roary process was repeated at 90% protein identity for each of the following groups: DT104 strains, non-DT104 strains, bovine-associated strains, and clinical strains. Identification of those genes unique to DT104 strains compared to the other ST19 non-DT104 strains were identified using the output of Roary as input into the program Scoary (v1.6.16) [59], and the analysis was run with default parameters and traits file containing the phage type for each of the strains (Scoary command: scoary -g gene_presence_absence.csv -t DT104_traits.csv). The Scoary analysis was repeated to identify genes unique to non-DT104 strains, bovine strains, and clinical strains among the thirteen ST19 *S. typhimurium* strains used in this study. An expanded DT104 cattle-associated unique gene analysis was conducted by selecting the 2009 Colorado outbreak ground beef strain RM9437, 25 cattle-associated DT104 strains from EnteroBase, and an additional 72 random DT104 genomes including at least one representative from each source of isolation from each available country in EnteroBase (Appendix A; excluding RM9435 and RM9436). The remaining 50 strains were random selected and downloaded from the list of 556 DT104 strains in Enterobase. The 98 strains were annotated with Prokka and analyzed via Roary at 90% protein identity as described above. To determine the cattle-specific DT104 unique genes, the output of Roary was input into Scoary as described above with the traits file containing cattle/bovine/beef versus other for each DT104 strain.

### 4.9. Pangenome Analysis and Visualization with Anvi’o Software

The 100 DT104 strains described above were also used to visualize the pangenome by analysis with Anvi’o software (v6.2) [60] using the pangenome workflow [61]. Pangenome analysis was conducted using NCBI’s blastp program with a minimum bit score 0.5 for gene clustering and using the MCL algorithm [62] to identify gene clusters. The gene clusters determined in the analysis were binned based on the following parameters: (1) core genes–clusters present in 100% of the genomes; (2) accessory genes–gene clusters present in ≤99% of the genomes but more than one genome; (3) singleton genes–gene clusters present in only one genome maximum. The country and source of isolation were imported as additional layers in the database.

## Figures and Tables

**Figure 1 pathogens-10-00529-f001:**
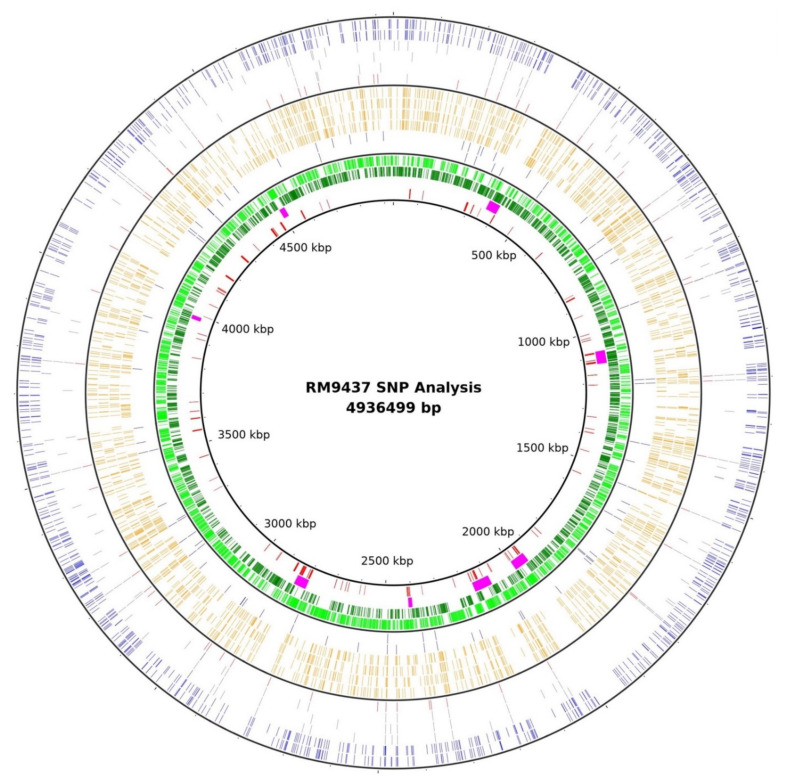
SNP map between ST19 *S. typhimurium* strains. Location of SNPs from different bovine and clinical *S. typhimurium* strains compared to RM9437. Beginning in the inner circle and moving outwards: (1) RM9437 repeat regions; (2) RM9437 prophage regions; (3) RM9437 forward CDSs; (4) RM9437 reverse CDSs; bovine strains: (5) SA972816 (DT104); (6) USMARC-1808 (DT104); (7) USMARC-1810; (8) USMARC-1880; (9) USMARC-1896; (10) USMARC-1898; human clinical strains: (11) DT104 (DT104); (12) CDC H2662 (DT104); (13) CDC 2009K-1640 (DT104); (14) CDC 2011K-1702; (15) CDC 2009K-2059; (16) CDC 2010K-1587.

**Figure 2 pathogens-10-00529-f002:**
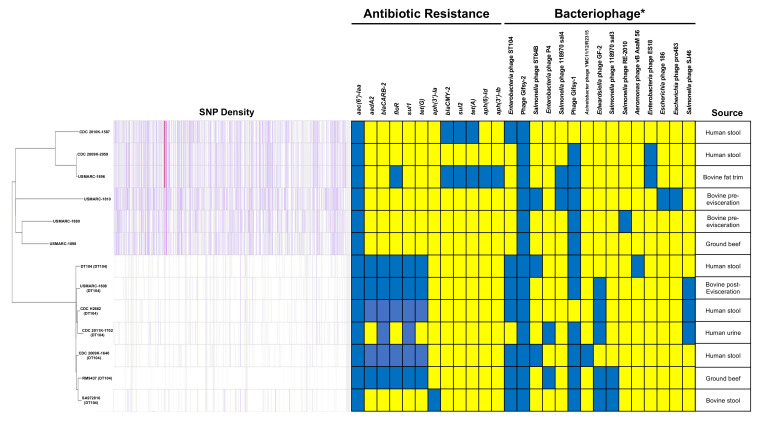
Phylogenetic relationship of *S. typhimurium* strains. Maximum-likelihood (ML) tree of all the bovine and clinical strains used in the study based on 1798 SNPs identified between the 13 strains by Parsnp software using *S. typhimurium* str. RM9437 as the reference strain. ML tree generated with MEGAX software using Tamura–Nei model with a discrete Gamma distribution with 1000 pseudoreplicates. Antibiotic resistance gene and prophage profiles for each of the strains were identified with ResFinder and PHASTER, respectively (blue—present; yellow—absent). Note: * Includes only bacteriophage identified as intact by PHASTER.

**Figure 3 pathogens-10-00529-f003:**
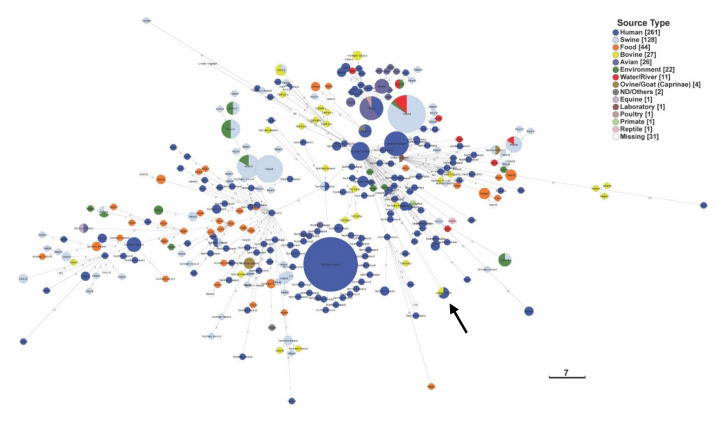
cgMLST analysis of *S. typhimurium* DT104 strains. Core genome multilocus sequencing type (cgMLST) minimum spanning tree (MST) based on 2978 alleles of 556 DT104 strains representing 330 sequence types (STs) present in Enterobase on 11 February 2021. The numbers represent number of alleles between strains, and the black arrows mark the three United States strains sequenced in this study. Nodes are colored according to source type, while grey nodes (ND/others) and white nodes (missing) had no source type metadata in Enterobase. Logarithmic scale represents number of alleles.

**Figure 4 pathogens-10-00529-f004:**
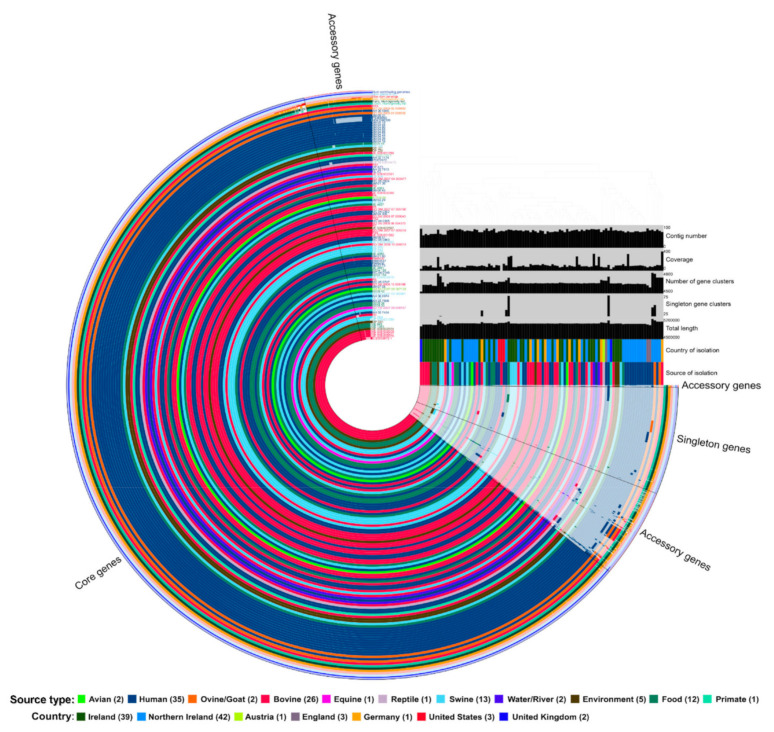
Pangenome of 100 DT104 strains. Visualization of pangenome using Anvi’o software for 100 DT104 strains from Enterobase including the 3 Colorado outbreak strains (RM9435, RM9436, and RM9437), 25 bovine-associated strains, and 72 random selected strains from other sources. Partial rings represent comparison of DT104 genomes, including core genes present in all 100 strains, accessory genes present in ≥2 strains but ≤99 strains, and singleton genes that are only present in a single DT104 genome. Color of the ring is based on the source of isolation/source type. Order of the rings from outer most to inner most: (1) number of DT104 genomes contributing to gene cluster marked in strain ring; (2) number of genes in the gene cluster; (3) maximum number of paralogs in gene cluster; (3) geometric homogeneity identity of gene cluster; (4) functional homogeneity identity of gene cluster; (5) combined homogeneity identity of gene cluster; *S. typhimurium* strains: (6) R13; (7) BD-DU-2006-02-006862; (8) AH-00-5945; (9) BD-DU-2006-01-008520; (10) JM-08-31; (11) P5368550; (12) H041040309; (13) JM-05-18; (14) JM-04-98; (15) JM-04-96; (16) JM-04-88; (17) JM-04-66; (18) JM-04-33; (19) JM-04-32; (20) JM-04-26; (21) JM-04-16; (22) SE04-41; (23) DP-J28; (24) DP-J21; (25) DP-J20; (26) JE-S09-001058; (27) S-20031116-1; (28) AH-02-7174; (29) P5372472; (30) CK-95-00014475; (31) R11; (32) DP-A73; (33) AH-03-7513; (34) NL-A3; (35) JE-S09-002301; (36) DP-A58; (37) BD-OM-2007-04-003477; (38) MC-05-0834; (39) JM-01-36; (40) R9; (41) JE-4963; (42) JM-06-45; (43) JE-S09-002300; (44) R4; (45) A1519-04; (46) JM-02-29; (47) DP-A68; (48) JE-4657; (49) R5; (50) BD-OM-2007-07-005180; (51) MC-04-0302; (52) JM-04-300; (53) BD-DU-2006-07-000043; (54) R12; (55) MC-04-0395; (56) BD-DU-2006-06-004373; (57) R7; (58) JE-S09-002567; (59) BD-OM-2007-07-005216; (60) R10; (61) JE-S09-001062; (62) JM-08-01; (63) MC-05-0863; (64) JE-S09-002695; (65) BD-OM-2006-10-009515; (66) DP-A60; (67) DP-A43; (68) DP-A59; (69) JE-4960; (70) JM-01-80; (71) RM9435; (72) RM09437; (73) RM9436; (74) JM-03-73; (75) JE-3481; (76) SE04-155; (77) MC-04-0146; (78) SE04-154; (79) JE-S09-000879; (80) R8; (81) MC-05-0747; (82) BD-DU-2006-10-005166; (83) JM-07-08; (84) SM-DU-2007-09-007135; (85) SE06-42; (86) SM-DU-2007-03-009861; (87) AH-06-8974; (88) NL-I12; (89) AH-03-7508; (90) SE06-34; (91) SE06-33; (92) BD-PM-2007-09-006137; (93) NL-I11; (94) AH-03-7454; (95) R3; (96) NL-R64; (97) JE-S09-001059; (98) DP-D21; (99) JE-843; (100) JE-1363; (101) JE-S09-002574; (102) JE-S09-002079; (103) JE-S09-002077; (104) JE-S09-002078; (105) S-20030672-1.

**Figure 5 pathogens-10-00529-f005:**
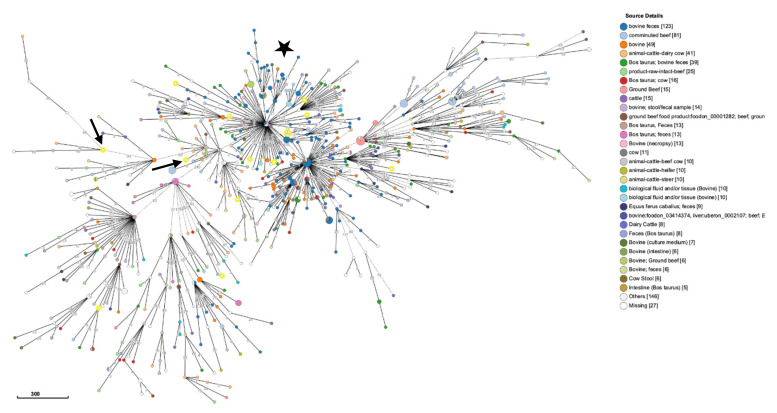
cgMLST analysis of North American *S. typhimurium* bovine strains from ST19 lineage. Core genome multilocus sequence type (cgMLST) minimum spanning tree (MST) based on 2978 alleles of 775 *S. typhimurium* strains isolated from different bovine sources in North America that belong to the ST19 lineage and represent 682 core genome sequence types (ST) present in Enterobase on 14 May 2020. The numbers represent number of alleles between strains, and the fifteen strains used in this study are highlighted in yellow. Nodes are colored according to source details, and white nodes had the metadata missing. The arrows represent the two non-DT104 clinical strains, whereas the star marks the node with all seven DT104 strains. Logarithmic scale represents number of alleles.

**Table 1 pathogens-10-00529-t001:** *Salmonella typhimurium* strains used in this study.

Strain	Chromosome Size (bp)	Number of Plasmids	Prophage *	DT104	Collection Year	Country of Isolation	Isolation Source	Accession Number	Reference
RM9437	4,936,499	1	8 (6)	Yes	2009	USA: Colorado	Ground Beef	CP012985	This study
RM9436	4,936,499	1	8 (6)	Yes	2009	USA: Colorado	Human Stool	SRR8660931	This study
RM9435	4,936,499	1	8 (6)	Yes	2009	USA: Colorado	Human Stool	SRR8660930	This study
CDC H2662	4,891,165	1	6 (4)	Yes	1997	USA	Human Stool	CP014979	[26]
CDC 2009K-1640	4,933,707	1	8 (5)	Yes	2009	USA: BIFSCo Region 2	Human Stool	CP014975	[26]
SA972816	4,891,923	2	8 (5)	Yes	2002	China	Bovine Stool	CP007484	[27]
CDC 2011K-1702	4,906,321	1	10 (5)	Yes	2011	USA: BIFSCo Region 7	Human Urine	CP014967	[26]
USMARC-1808	4,936,894	1	7 (5)	Yes	2005	USA: BIFSCo Region 8	Bovine Post-evisceration	CP014969	[26]
DT104	4,933,631	1	9 (5)	Yes	1988	England	Human stool	HF937208.1	[22]
CDC 2009K-2059	4,823,793	0	6 (3)	No	2009	USA: BIFSCo Region 2	Human Stool	CP014983	[26]
CDC 2010K-1587	4,799,398	4	6 (2)	No	2010	USA: BIFSCo Region 8	Human Stool	CP014965	[26]
USMARC-1810	4,927,145	0	9 (6)	No	2005	USA: BIFSCo Region 5	Bovine Pre-evisceration	CP014982	[26]
USMARC-1880	4,815,208	0	6 (3)	No	2003	USA: BIFSCo Region 5	Bovine Pre-evisceration	CP014981	[26]
USMARC-1896	4,856,440	1	7 (4)	No	2011	USA: BIFSCo Region 2	Bovine fat trim	CP014977	[26]
USMARC-1898	4,784,385	3	5 (2)	No	2007	USA: BIFSCo Region 3	Ground Beef	CP014971	[26]

* Number in parentheses represent number of intact prophage sequences.

**Table 2 pathogens-10-00529-t002:** Total number of SNPs between different strains of *S. typhimurium* from human and bovine sources.

	RM9437	CDC 2009K-1640	CDC 2011K-1702	CDC H2662	SA972816	USMARC-1808	DT104	USMARC-1810	USMARC-1880	USMARC-1896	USMARC-1898	CDC 2009K-2059	CDC 2010K-1587
RM9437		54	85	60	30	67	61	694	593	627	584	617	650
CDC 2009K-1640	54		71	48	36	53	47	682	579	615	570	605	638
CDC 2011K-1702	85	71		51	67	68	72	709	606	642	599	632	665
CDC H2662	60	48	51		44	45	49	686	583	619	576	609	642
SA972816	30	36	67	44		49	43	678	575	611	566	601	634
USMARC-1808	67	53	68	45	49		54	691	588	624	581	614	647
DT104	61	47	72	49	43	54		681	578	614	571	604	637
USMARC-1810	694	682	709	686	678	691	681		591	631	584	621	646
USMARC-1880	593	579	606	583	575	588	578	591		512	367	502	529
USMARC-1896	627	615	642	619	611	624	614	631	512		505	10	305
USMARC-1898	584	570	599	576	566	581	571	584	367	505		495	524
CDC 2009K-2059	617	605	632	609	601	614	604	621	502	10	495		295
CDC 2010K-1587	650	638	665	642	634	647	637	646	529	305	524	295	

**Table 3 pathogens-10-00529-t003:** Cattle-associated *S. typhimurium* DT104 genes *.

Prokka Annotation	% of Bovine-Associated DT104 Genomes with Gene	% of Non-Bovine-Associated DT104 Genomes ^#^ with Gene	*p*-Value
Hypothetical protein	0%	33.3%	0.0003
Beta-lactamase PSE-1 (*pse1*)	100%	75.0%	0.0026
Regulatory protein Rop	3.8%	33.3%	0.0031
Hypothetical protein	3.8%	33.3%	0.0031
Hypothetical protein	3.8%	31.9%	0.0032
Hypothetical protein	3.8%	31.9%	0.0032
tRNA-Ala(tgc)	19.2%	1.4%	0.0047
Hypothetical protein	3.8%	30.5%	0.0059
Mobilization protein MbeC	3.8%	30.5%	0.0059
Endoribonuclease ToxN	3.8%	30.5%	0.0059
Dihydropteroate synthase	3.8%	29.2%	0.0062

* Genes listed are those statistically significant (*p*-value < 0.01). ^#^ Non-bovine sources include avian, clinical, ovine, equine, reptile, swine, water, environment, food, and primate. Note: 2009 Colorado outbreak strains RM9435 and RM9436 were not used in the analysis as these clinical strains are identical to the ground beef strain RM9437 and from the same outbreak.

## Data Availability

The sequence data presented or used in this study are openly available at either NCBI Genbank under the accession numbers listed in Table 1 or Enterobase as listed in Appendix A.

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
