# Peer review of "Genomic Characterization of Salmonella typhimurium DT104 Strains Associated with Cattle and Beef Products"

_pathogens, 2021, doi:10.3390/pathogens10050529_

Round 1
Reviewer 1 Report
This is an extensive largely bioinformatic study which provides insights into S. Typhimurium DT104 in beef cattle and on beef products. The methods used are appropriate and the analysis extensive. The conclusions drawn are useful and not overstated.
There are some very minor issues with the language and editing. I provide some examples below but this list is not comprehensive. I suggest the authors revise the manuscript very thoroughly as clarity is lost in a number of these cases. In particular, some sentences are very long and require re-reading a number of times to extract their meaning.
Line 114: "quite a bit" is not an appropriate scientific term.
Line 169: Sentence begins "Including a partial deletion..." does not make sense as it stands.
Line 209: Add "which were" between "strains" and "distributed".
Line 218: "suggesting" should be "we suggest"?
Line 232: "prospective" should be "perspective"?
Line 238-242: Example of a long sentence which would be easier to understand if it was two sentences.
Reviewer 2 Report
This manuscript describes the whole genome sequencing of three Salmonella DT104 isolates and the resulting comparison of genome sequences from DT104 and non-DT104 clinical and bovine isolates using data available from NCBI sources. Both DT104 and non-DT104 isolates were isolated from humans and cattle, and the genomic diversity of non-DT104 isolates was much greater than DT104 isolates. These data will fill a gap in surveillance, since few sequences from North American Salmonella DT104 isolates are currently available.
Additional questions and comments:
- lines 16-18 (Abstract), “…we sequenced several S. Typhimurium DT104 strains associated with a 2009 outbreak caused by ground beef, including the outbreak source strain and 17 several clinical strains.”
Table 1 lists all strains with complete sequences used in the study. Only three were from this study according to the Reference column, and others were from other studies. It is not clear which were sequenced as part of this study. It must be assumed that the three from this study were the outbreak strains. Were the others as well? Please clarify this and indicate which isolates were sequenced for this study in the table and in the Materials and Methods. If only three isolates were sequenced for the study, please state that in the Abstract.
- Table 1 lists references 26-28 as the source of data for several of the strains. However, these references do not appear to be correct. Could you please check this?
- p. 5, lines 7-9, “…the 7 DT104 strains had significantly less SNPs compared to the non-DT104 strains (Figure 1).”
Would it be possible to provide the state in which US isolates were obtained so that the reader can see the geographic distribution of strains as an aid to interpretation?
- p. 8, lines 14-17, “…38 genes were present in the Aeromonas phage vB AsaM-56 (as identified by PHASTER analysis) and the other genes were present in the Salmonella phage ST64B (Supplemental Table 4), but these genes were part of other prophages in the other DT104 strains.”
It is not clear to me exactly what you mean here. You do not mention that these prophages were intact in only one DT104 strain, which could suggest that they were present intact in other strains. But the statement that the genes were part of other prophages (different prophages?) in other DT104 strains negates that assumption. Please clarify. If the genes unique to DT104 were present in different prophages, were they clustered the same way? Are these prophages variants in a manner similar to some STEC prophages? This may be worth following up in future analysis.
- Table 3. Cattle associated S. Typhimurium DT104 genes that are present/absent compared to other sources*.
I did not find a reference to Table 3 in the body of the text. Such a reference should be added, with explanation and interpretation, or the Table should be removed.
The meaning of this title is not clear. What are the other sources?
- p 3 of 23, lines 71-74, “Only two genes were statistically more commonly present in cattle DT104 strains than non-cattle associated DT104 strains. The beta-lactamase PSE-1 (pse1) gene, which was present in all 26 bovine DT104 strains examined, but only present in 75.0% of the non-cattle associated DT104 strains (p value = 0.0026).”
Despite the decent p value, this seems like a pretty weak difference. How many non-cattle-associated strains were included? Could this finding be skewed a bit by the sampling?
- Fig 5.
I could not see the arrows or the star. It might just be me, but please check this.
- cgMLST analysis.
Has the schema for the cgMLST been described elsewhere? If not, it would be very good to make a note of how this was developed in the Materials and Methods.
Reviewer 3 Report
The manuscript of C.T. Parker et al. "Genomic characterization of Salmonella Typhimurium DT104 strains associated with cattle and beef products" is devoted to the comparative genomic study of Salmonella Typhimurium DT104 isolated from the 2009 Colorado outbreak and comparison with other DT104 and non-DT104 S. Typhimurium genomes. The authors performed whole-genome sequencing of three strains, one from ground beef and two isolated from patients.
Although I did not found major issues with the manuscript, I present my minor issues and suggestions below, which help to improve the text.
Line 4: Omit "#"
Line 10: Omit repeated "correspondence to"
Lines 11-12: No need to put the address for the corresponding author, just E-mail (line 10) is enough.
Lines 16 and 18: Put three and two strains, respectively, for total and clinical strains from the outbreak.
Line 46: Reference is too outdated. It was true in the 2000s, but not for the 2010s. Currently, S. Enteritidis is the most common serovar worldwide, especially for humans, and S. Typhimurium is the most common one for animals.
Line 102: Again, put "two clinical isolates".
Line 111: Remove dot between 2009 and Colorado.
Line 114: Could you describe more details for 92,355 bp plasmid: does it contains virulence, eg spv-operon, or AMR genes? Moreover, could you explain why in GenBank deposited plasmid pRM9437 size is different, 93,930 bp?
Line 121: You can omit genes within brackets.
The enumeration and page numbering is broken, so each page started from 1, sorry.
pg. 8, line 10: Supplemental Table 3 before 1 and 2 (in Materials and Methods)?
Table 3: Typo - should be as "(p-value < 0.01)"
pg. 12, line 70: 10 genes in the text, but 11 in Table 3.
pg. 12, lines 70 and 71: What is the difference between "statistically significant" and "statistically more commonly"?
pg. 12, line 78: Probably, you want to say not proteins, but genes encoding proteins...
Figure 3: Poor resolution: unreadable text.
Figure 3: What is the difference between ND/Others and Missing?
Figure 3, line 83: replace ST with STs
Figures 3 and 5: I can't find any black arrows?
Figure 3: I can't distinguish "numbers represent number of allele differences between strains". Completely unreadable.
Figure 3, line 83 and also in Figure 5 and Materials and Methods: probably not "alleles", but "loci" or "genes"?
Figures 4 and 5: Again, poor resolution and unreadable text.
Figure 5: I can't find star marks.
pg. 14, line 131: S. Typhimurium
pg. 15, line 154: replace "serotypes" with "serovars" because you used it above.
pg. 16, line 210: MST instead of minimum spanning tree
pg. 16, line 235: ST instead of sequence type
pg. 17, lines 264-265: Models of Qubit and Nanodrop and wavelength used.
pg. 17, line 286: "de novo" should be not italicized
pg. 17, line 288: manually ordered?
pg. 18, line 303: Omit "automatic", it is just PGAP
pg. 19, line 351: version of ResFinder?
pg. 19, line 358: MST
pg. 19, line 361: remove "sequence types"
pg. 19, line 385: Could you describe here how randomization was performed?
Round 2
Reviewer 2 Report
The authors have answered all questions and dealt with all issues identified in the initial review.